# Effects of Weight Status and Related Metabolic Disorders on Fertility-Sparing Treatment Outcomes in Endometrial Atypical Hyperplasia and Endometrial Cancer: A Retrospective Study

**DOI:** 10.3390/cancers14205024

**Published:** 2022-10-14

**Authors:** Sijia Liu, Lulu Wang, Pengfei Wu, Shuhan Luo, Weiwei Shan, Xiaojun Chen, Xuezhen Luo

**Affiliations:** 1Department of Gynecology, Obstetrics and Gynecology Hospital of Fudan University, Shanghai 200011, China; 2Shanghai Key Laboratory of Female Reproductive Endocrine Related Diseases, Shanghai 200011, China

**Keywords:** endometrial cancer, endometrial atypical hyperplasia, metabolic disorder, hyperuricemia, underweight

## Abstract

**Simple Summary:**

Fertility-sparing treatment for young women with endometrial atypical hyperplasia or endometrioid endometrial cancer has become an important priority. The aim of our retrospective study was to evaluate the effects of different weight statuses and related metabolic disorders on the oncological and reproductive outcomes of fertility-sparing treatment. We found that there was a parabola-shaped relationship between the cumulative complete response rate of treatment and BMI. The apex of the curve was observed at a BMI of 21–22 kg/m^2^. Furthermore, we demonstrated that hyperuricemia was an independent risk factor for the failure of conservative treatment, correlating with a lower cumulative 32-week CR rate and longer treatment duration. Our results indicate that a target BMI interval for weight management should be appropriately established for patients with EAH/EEC. Early active interventions for related metabolic disorders, preferably before 32 weeks of treatment, should be provided to improve treatment efficacy.

**Abstract:**

Background: Although obesity was an independent risk factor for fertility-sparing treatment in endometrial atypical hyperplasia (EAH) and endometrioid endometrial cancer (EEC), the roles of other weight statuses and related metabolism were unclear. This study aimed to investigate the body mass index (BMI) interval that produced optimal treatment efficacy and the effects of related metabolic disorders in EAH/EEC patients. Methods: A total of 286 patients (including 209 EAH and 77 well-differentiated EEC) under progestin therapy were retrospectively analyzed. The cumulative complete response (CR) rate, relapse rate, and fertility outcomes were compared among different weight or metabolic statuses. Results: Underweight and overweight/obese status significantly decreased the cumulative 16-week and 32-week CR rate (*p* = 0.004, *p* = 0.022, respectively). The highest 16-week CR rate was observed at a BMI of 21–22 kg/m^2^ in the overall population (*p* = 0.033). Obesity (HR 0.37, 95%CI 0.15–0.90, *p* = 0.029) and PCOS (HR 0.55, 95%CI 0.31–0.99, *p* = 0.047) were associated with lower 16-week CR rate. Hyperuricemia (HR 0.66, 95%CI 0.45–0.99, *p* = 0.043) was associated with lower 32-week CR rate. The 16-week and 32-week CR rate (*p* = 0.036, *p* = 0.008, respectively) were significantly lower in patients exhibiting both obesity and hyperuricemia. Conclusions: The optimal fertility-sparing treatment efficacy was observed at a BMI of 21–22 kg/m^2^ in EAH/EEC. Hyperuricemia was an independent risk factor for long-term treatment outcomes.

## 1. Introduction

Endometrial cancer (EC) is one of the most common gynecological malignancies in women worldwide, with an increasing trend of incidence [1,2]. A substantial body of evidence suggests that EC and its precursor, endometrial atypical hyperplasia (EAH), are closely correlated with obesity and metabolic disorders, including diabetes, hyperlipidemia, and hypertension [3,4,5,6]. Observations linking other obesity-related metabolic factors, such as hyperuricemia (HUA), to EC/EAH have provided far less conclusive results [7].

Approximately 7% of EC cases are reported in premenopausal women under 45 years of age, 70% of which are nulliparous at the time of diagnosis [8]. Fertility-sparing treatment is preferred over total hysterectomy for young patients with strong fertility needs [9]. For EAH or well-differentiated endometrioid endometrial cancer (EEC), high-dose progestin has been widely accepted as the main fertility-sparing treatment.

Hysteroscopic resection combined with hormone therapy was also reported to be an effective fertility-sparing treatment [10]. Negative oncological and reproductive treatment outcomes have been strongly associated with obesity [11,12]. However, the role of underweight status in fertility-sparing treatment remains unknown. Several studies suggested that metabolic disorders, including insulin resistance (IR), polycystic ovary syndrome (PCOS), and metabolic syndrome (MetS), have adverse effects on progestin therapy [13,14,15]. Whether other obesity-related metabolic factors, such as lipid or uric acid (UA) metabolism, affect fertility-sparing treatment for EAH/EEC needs further study.

In this single-center retrospective study, we aimed to evaluate the effects of different weight statuses and related metabolic disorders on the oncological and reproductive outcomes of fertility-sparing treatment in EAH/EEC to ultimately provide better clinical guidance.

## 2. Materials and Methods

### 2.1. Study Population

Overall, 472 patients (329 with EAH and 143 with EEC) who accepted fertility-sparing treatment at the Obstetrics and Gynecology Hospital of Fudan University from January 2017 to August 2019 were retrospectively analyzed. Following the World Health Organization (WHO) pathological classification (2014), the diagnosis of EEC or EAH (equivalent to endometrial intraepithelial neoplasia, which was utilized more frequently in WHO 2020) was confirmed by two experienced gynecological pathologists. Both of them had extensive practice experience in the field of gynecologic tumor pathology diagnosis and had title of associate chief physician or above. If their opinions differed, a discussion would be held in the pathology department for a final diagnosis.

The criteria for inclusion were in strict accordance with the National Comprehensive Cancer Network guidelines as follows: (1) pathologically confirmed EAH or well-differentiated (grade 1) EEC on dilation and curettage (D&C) with or without hysteroscopy; (2) disease limited to the endometrium as observed by enhanced magnetic resonance imaging or transvaginal ultrasound; (3) no suspicious or metastatic lesions on imaging; (4) no contraindications to medical treatment or pregnancy; (5) non-pregnant state; (6) aged between 18–45 years old. The criteria for exclusion were as follows: (1) local or systemic progestin treatment history for more than one month before the initial evaluation and treatment in our center; (2) recurrent EAH or EEC; (3) transferred to other hospitals or required surgery before the first hysteroscopy to assess treatment efficacy; (4) incomplete necessary medical records.

This study was approved by the Ethics Committee of the Obstetrics and Gynecology Hospital of Fudan University. All patients received comprehensive information about the risks of surgery and fertility-sparing treatment. All patients signed informed consent for conservative therapy and the use of their clinical data for research purposes.

### 2.2. Diagnosis and Assessment

The general information was collected, including age, height, weight, waist circumference, and medical complications. The serum samples were collected before any treatment and analyzed for glucose, insulin, lipid profile, sex hormone profile, anti-Mullerian hormone (AMH), and UA levels. All of the samples were examined in the laboratory of the Obstetrics and Gynecology Hospital. The tests were repeated if the results exceeded the normal range.

The weight status was classified by the ethnic-specific BMI cut-off points recommended by the WHO. Underweight was defined as a BMI less than 18.5 kg/m^2^. Normal weight was defined as a BMI greater than 18.5 kg/m^2^ and less than 25.0 kg/m^2^. Overweight was defined as a BMI greater than 25.0 kg/m^2^ and less than 30.0 kg/m^2^. Obesity was defined as a BMI greater than 30.0 kg/m^2^. According to our previous study [16], we considered patients had IR status when the homeostasis model assessment-insulin resistance was ≥2.95. The diagnoses of hyperlipidemia, hypercholesterolemia, hypertriglyceridemia, hypo-high-density lipoprotein cholesterolemia (hypo-HDL), and hyper-low-density lipoprotein cholesterolemia were following the Chinese adult dyslipidemia prevention guide (2016) [17]. A diminished ovarian reserve was defined as having at least one of the following criteria: (1) serum AMH levels less than 1.1 ng/mL; (2) antral follicular count less than 7 follicles [18]. Diagnosis of PCOS was based on Rotterdam Consensus Criteria [19]. Hyperuricemia was defined as fasting serum UA concentrations greater than 357 μmol/L on two different days [20]. The diagnostic criteria for MetS were described in our previous article [13].

### 2.3. Treatment and Evaluation

All of the patients received one of the following progestin therapies: (1) oral megestrol acetate (MA) at a dose of 160 mg/d; (2) levonorgestrel intrauterine system (LNG-IUS) insertion; (3) MA combined with LNG-IUS. Some patients also received oral metformin at a dose of 1500 mg/d, depending on medical complications. During the fertility-sparing treatment, all patients were actively managed in order to keep track of their medication adherence and adverse effects.

Hysteroscopy was performed every 3 months to evaluate the fertility-sparing treatment efficacy, following the standard procedure as described previously [10]. We extended the cut-off points for analysis to the 16th and 32nd week due to the slight advance or delay in the date of patients’ hysteroscopic evaluations. Endometrial lesions were removed under hysteroscopy, and if no obvious lesions were found, the endometrium was randomly biopsied. The specimens were sent for pathological examination. Complete response (CR) was defined as an absence of hyperplasia or cancer. The pathology of endometrial specimens could be secretory/proliferative endometrium after treatment. Partial response was defined as a pathological improvement, i.e., the presence of complex/simple hyperplasia lesions in endometrial specimens of EAH patients after treatment or atypical/complex/simple hyperplasia lesions in EEC patients after treatment. Disease progression was defined as the possibility of EC in EAH or the presence of myometrial infiltration or extra-uterine metastasis in patients with EEC.

Patients with stable disease over 6 months, partial response over 9 months, or disease progression at any time during the treatment would be strongly recommended for hysterectomy. Alternative treatments would be provided for patients who refused surgery according to recommendations from the multidisciplinary team.

### 2.4. Fertility and Follow-Up

Consolidation therapy with the original program for 3 months was performed for patients who achieved CR. If CR was confirmed by the next hysteroscopy evaluation, the patients were advised to prepare for pregnancy or assisted reproductive technology as soon as possible. Low-dose progesterone, oral contraceptives, or LNG-IUS were recommended to prevent recurrence for patients who were not pregnant or had not given birth. All of the patients were followed up until August 2020.

### 2.5. Statistics Analysis

The categorical variables were presented as frequency and percentage. The descriptive variables were presented as mean and standard deviation or median and interquartile range (IQR). We used the Chi-square test or Fisher’s exact test to analyze the differences in the categorical variables where appropriate. The *p*-value for pairwise comparison was adjusted by the Bonferroni methods. The differences in the descriptive variables between the two groups were analyzed by the Student’s *t*-test or Mann–Whitney U test, and differences among more than two groups were analyzed by one-way ANOVA or the Kruskal–Wallis H test where appropriate. A logistic regression model was used to analyze the correlation between weight status and metabolic disorders in patients with EAH/EEC. The Kaplan–Meier method was used to evaluate the cumulative CR rate or relapse rate. The log-rank test was used for comparisons among groups. Restricted cubic spline with five knots at the 5th, 35th, 50th, 65th, and 95th centiles were used to model the non-linear association between BMI and therapeutic efficacy. The Cox regression model was used to analyze the effects of variables on fertility-sparing treatment outcomes. A *p*-value < 0.05 (two-tailed) was considered statistically significant. All statistical analyses were performed in SPSS (version 23.0, IBM, Armonk, NY, USA).

## 3. Results

A total of 472 patients with EAH/EEC receiving fertility-sparing treatment in the Obstetrics and Gynecology Hospital of Fudan University from January 2017 to August 2019 were retrospectively investigated (Figure 1). Overall, 186 cases were excluded, including 88 cases who had accepted progestin therapy for more than one month before the first endometrial assessment at our center. We did not analyze these patients because of the potential effects of their previous treatment (including cyclic progestin, continued progestin, oral contraceptive pills, or LNG-IUS), particularly those who had poorer outcomes in initial hospitals. Before the first hysteroscopy was performed to assess the conservative efficacy, 42 cases requested surgical treatment, and 26 cases who requested a transfer to another hospital were excluded. In addition, 19 cases with recurrent EAH or EEC and 11 cases with incomplete clinical data were also excluded. Ultimately, 286 patients who met all of the inclusion and exclusion criteria were retrospectively analyzed in this study, including 209 patients with EAH (73.1%) and 77 patients with EEC (26.9%). The median age at diagnosis was 32 years (IQR, 29–36 years), and the median BMI was 24.9 kg/m^2^ (IQR, 21.8–28.7 kg/m^2^). Overall, 78.7% of patients (225/286) were nulliparous at diagnosis. The median follow-up time of this cohort was 19.1 months (IQR, 12.1–27.8 months). As of the last follow-up, no patients were found to have poor medication adherence. 90.2% of patients (258/286) achieved CR with a median treatment duration of 6.2 months (IQR, 3.6–8.7 months).

### 3.1. Correlation between Weight Status and Metabolic Disorders

The study cohort included 14, 135, 85, and 52 cases that were underweight (4.9%), normal weight (47.2%), overweight (29.7%), and obese (18.2%), respectively. As shown in Table 1, overweight/obese patients were more likely to have a history of diabetes (*p* < 0.001), IR (*p* < 0.001), hypertriglyceridemia (*p* < 0.001), MetS (*p* < 0.001), hypertension (*p* < 0.001), PCOS (*p* = 0.005), and HUA (*p* = 0.001). No significant differences were noted across weight status in age at diagnosis (*p* = 0.068), histology (*p* = 0.951), progestin therapy (*p* = 0.128), and follow-up time (*p* = 0.222). The correlation between weight status and metabolic disorders in patients with EAH/EEC was further evaluated by a logistic regression model adjusted for age at diagnosis and histology (Table 2). Compared with normal weight patients, the overweight patients were prone to have diabetes [odds ratio (OR) 5.79, 95% confidence interval (CI) 2.01–16.74, *p* = 0.001], IR (OR 9.94, 95%CI 5.10–19.35, *p* < 0.001), hypertriglyceridemia (OR 6.37, 95%CI 2.50–16.20, *p* < 0.001), MetS (OR 12.23, 95%CI 6.19–24.15, *p* < 0.001), and hypertension (OR 3.19, 95%CI 1.75–5.79, *p* < 0.001). Obese patients had a higher risk of PCOS (OR 2.89, 95%CI 1.44–5.82, *p* = 0.003) and HUA (OR 3.06, 95%CI 1.50–6.25, *p* = 0.002) in addition to the above-mentioned metabolic disorders. Notably, the underweight patients were more likely to show hypo-HDL (OR 3.50, 95%CI 1.02–11.97, *p* = 0.047) than the normal weight group.

### 3.2. Effects of Weight Status on Oncological and Reproductive Treatment Outcomes

Table 3 summarizes the fertility-sparing treatment outcomes across patients with different weight statuses. Compared with the normal-weight patients, the cumulative 16-week CR rate was significantly lower in the overweight, obese, and underweight patients (34.1% vs. 23.5%, 11.6%, and 7.1%, respectively, *p* = 0.004). Similar results were observed at the 32nd week of treatment (67.1% vs. 59.7%, 48.5%, and 53.6%, respectively, *p* = 0.022) (Figure 2A). The median time from initiating treatment to CR in patients of normal weight was 5.1 months (IQR, 3.3–7.9 months). Abnormal weight status also significantly prolonged treatment duration (*p* = 0.015). The cumulative 24-month CR rate of the overall population was 98.2%, and no statistical difference was observed (*p* = 0.098) between patients with different weight statuses.

We performed a restricted cubic spline based on a Cox regression model to further evaluate the favorable BMI interval that produced optimal treatment efficacy. As shown in Figure 3A, BMI was strongly correlated with the cumulative 16-week and 32-week CR rate in the overall population (*p* = 0.002 and *p* = 0.003, respectively) after adjusting for age at diagnosis, histology, and progestin therapy. There was a significant non-linear relationship between the 16-week CR rate and BMI (*p* = 0.033). The apex of the curve was observed at a BMI of approximately 21–22 kg/m^2^. Table A1 shows the hazard ratio (HR) and 95%CI of some selected BMI values. Compared with the reference (BMI of 21 kg/m^2^), the HR was 0.54 (95%CI, 0.24–1.22), 0.48 (95%CI, 0.25–0.95), and 0.27 (95%CI, 0.13–0.54) with a BMI of 18, 25, and 30 kg/m^2^, respectively. When BMI continued to increase above 30 kg/m^2^, the decrease in the CR rate tended to be flat. Similar results were found until the 32nd week of treatment. The HR was 0.72 (95%CI, 0.44–1.19), 0.51 (95%CI, 0.32–0.82), and 0.45 (95%CI, 0.30–0.69) with a BMI of 18, 25, and 30 kg/m^2^, respectively. The decreasing trend of the CR rate became flat when the BMI was above 25 kg/m^2^. In subgroup analysis, the highest CR rate in patients with EAH was observed at a BMI of 21–22 kg/m^2^. However, the non-linear relationship was not significant (Figure 3B). In the EEC subgroup, no non-linear relationship was found between BMI and the cumulative 16-week (*p* = 0.628) or 32-week CR rates (*p* = 0.600).

The cumulative 24-month relapsed rate was 29.3%, with a median time to relapse of 13.8 months (IQR, 8.6–19.8 months). Fourteen cases relapsed during ovulation induction therapy, six cases relapsed during maintenance therapy (four cases with Diane-35 and two cases with LNG-IUS), and one case relapsed because of irregular medication. No difference was observed in cumulative relapse rate (*p* = 0.418) or time to relapse (*p* = 0.796) across patients with different weight status. However, obese cases seemed to show a lower relapsed rate (7.3%) and a shorter time to relapse (6.3 ± 0.2 months). In pairwise comparison, the recurrence of obese patients occurred significantly sooner than normal-weight patients (6.3 months vs. 12.6 months, *p* = 0.031).

A total of 161 patients tried to conceive after achieving CR, and 36 cases (22.4%) had successful pregnancies. For these 161 patients, the miscarriage and live birth rates were 22.2% and 50.0%, respectively. Until the last follow-up, ten cases were still pregnant. No difference was observed in reproductive treatment outcomes. No serious adverse events related to drugs and hysteroscopies, such as thromboembolism, severe renal or hepatic dysfunction, serious infection, or uterine perforation, were observed.

### 3.3. Effects of Related Metabolic Factors on Treatment Outcomes

Table 4 shows the association between the related metabolic disorders and oncological treatment outcomes. Compared to non-IR or non-PCOS patients, the cumulative 16-week CR rate was significantly lower in patients with IR (19.0% vs. 29.5%, *p* = 0.041) or PCOS (15.4% vs. 31.6%, *p* = 0.003), which has been demonstrated in our previous study [13,14]. At the 32nd week of treatment, the cumulative CR rate was lower in patients with HUA or MetS than with non-HUA (48.6% vs. 64.9%, *p* = 0.009) or non-MetS (53.8% vs. 63.9%, *p* = 0.037) (Figure 2B,C). No differences were observed between the PCOS and non-PCOS patients (57.7% vs. 61.0%, *p* = 0.203) or the IR and non-IR patients (57.8% vs. 62.8%, *p* = 0.147). Furthermore, diabetes (6.0 vs. 7.1 months, *p* = 0.045), PCOS (5.4 vs. 6.9 months, *p* = 0.045), and HUA (5.8 vs. 7.1 months, *p* = 0.047) prolonged the treatment duration to achieve CR. No differences were found with the presence of other metabolic factors.

Univariate Cox regression analysis showed that obesity (HR 0.29, 95%CI 0.12–0.68, *p* = 0.004), IR (HR 0.59, 95%CI 0.35–0.99, *p* = 0.043), and PCOS (HR 0.44, 95%CI 0.25–0.77, *p* = 0.004) were associated with a lower cumulative 16-week CR rate. The adverse effects of obesity (HR 0.37, 95%CI 0.15–0.90, *p* = 0.029) and PCOS (HR 0.55, 95%CI 0.31–0.99, *p* = 0.047) were still significant in multivariate Cox analysis after adjusting for age at diagnosis, weight status, IR, and PCOS (Figure 4A). Furthermore, we found that the pathology type of EEC (HR 0.68, 95%CI 0.48–0.98, *p* = 0.038), obesity (HR 0.51, 95%CI 0.32–0.81, *p* = 0.004), HUA (HR 0.60, 95%CI 0.41–0.89, *p* = 0.010), and MetS (HR 0.72, 95%CI 0.62–0.98, *p* = 0.039) were correlated to a lower cumulative 32-week CR rate in univariate Cox analysis. Both HUA (HR 0.66, 95%CI 0.45–0.99, *p* = 0.043) and EEC (HR 0.69, 95%CI 0.48–0.99, *p* = 0.045) remained independent risk factors in multivariate Cox analysis after adjusting for histology, weight status, HUA, and MetS (Figure 4B).

Considering the close correlation between obesity and HUA, we next stratified obese and non-obese patients according to HUA status (Figure 2D). Significant differences were found among the four groups examined in the 16th week of treatment (*p* = 0.036). The non-obese patients with normal UA levels had the highest cumulative 16-week CR rate of 30.8%. For patients with HUA only (non-obese) or obesity only (normal UA), the CR rate was 20.8% and 13.1%, respectively. The cumulative CR rate was significantly lower (9.5%) in patients exhibiting both obesity and HUA. Similar results were found in the 32nd week of treatment (*p* = 0.008). Obesity and HUA may synergistically impact conservative therapy outcomes in patients with EAH/EEC.

## 4. Discussion

In this single-center retrospective analysis, we found that there was a parabola-shaped relationship between the cumulative CR rate of fertility-sparing treatment and BMI. The apex of the curve was observed at a BMI of 21–22 kg/m^2^. Furthermore, we demonstrated that HUA was an independent risk factor for the failure of conservative treatment, correlating with a lower cumulative 32-week CR rate and longer treatment duration.

### 4.1. Effects of Weight Status on Fertility-Sparing Treatment

A substantial body of evidence has shown that BMI plays an important role in the development and prognosis of EAH/EEC [3]. Multiple mechanisms were proposed to explain the risk correlation between overweight/obesity and EC, such as sex hormone metabolism, insulin and insulin-like growth factor signaling, and adipokine pathophysiology. Subclinical inflammation, closely related to the adipokine system, was considered one of the most important mechanisms [21,22,23]. In our study, obesity [BMI ≥ 30 kg/m^2^] was observed as an independent risk factor for the cumulative 16-week CR rate in patients with EAH/EEC. Several previous studies confirmed that overweight/obesity status had adverse effects on conservative treatment [11,13,14]. Elevated BMI was characterized by excess fat accumulation, which induced the hypersecretion of peripheral and local estrogens [24], abnormal function of endometrial stromal cells [25], or an inflammatory microenvironment in the endometrium [26], leading to progesterone resistance and poor progestin treatment outcomes.

We surprisingly found that underweight status (BMI < 18.5 kg/m^2^) also had an adverse impact on fertility-sparing treatment in patients with EAH and EEC, characterized by a lower cumulative CR rate and longer treatment duration. A significant non-linearity relationship was observed between the 16-week CR rate and BMI in all patients (*p* = 0.033). The CR rate reached an apex when the BMI was approximately 21–22 kg/m^2^. The incidence of endometrial diseases was low in underweight women, and no research to date has investigated the mechanism between underweight status and poor treatment outcomes. In the current study, underweight patients were more likely to exhibit hypo-HDL. Several studies suggested a close correlation between HDL metabolism and adiponectin [27,28], which exerted both anti-inflammatory and anti-proliferative actions in endocrine cancers [29]. Yamauchi et al. found that AdipoR1 and AdipoR2, the receptors of adiponectin, were significantly associated with progesterone receptor expression in EC [30]. The progesterone insensitivity mediated by low adiponectin levels may lead to poor outcomes of hormone therapy. A retrospective analysis from Japan found that young women with underweight status were more likely to show impaired glucose tolerance, which was characterized by impaired early insulin secretion and insulin insensitivity [31]. In a German study, patients with PCOS and underweight status had higher postprandial insulin levels [32]. The postprandial IR status may provide another possible explanation for the negative effects of underweight status on fertility-sparing treatment.

Gonthier et al. found that obese patients with EAH/EEC had a lower pregnancy rate after conservative treatment [11]. However, no significant differences in reproductive outcomes were observed across patients with different weight statuses in the present study. The active use of assisted reproductive technology for obese patients may be one of the possible reasons. Until the last follow-up, ten patients were still pregnant. Extending the follow-up period may be helpful in finding significant results.

### 4.2. Effects of Metabolic Disorders on Fertility-Sparing Treatment

EAH and EEC were closely associated with metabolic diseases, including MetS, diabetes, hyperlipidemia, and hypertension [6,33]. Our previous studies demonstrated that IR and PCOS were also risk factors for endometrial diseases [13,14]. In this study, patients with MetS had a lower 32-week CR rate than those with non-MetS. Li et al. indicated that a higher relapse rate was associated with MetS after conservative therapy in patients with EAH/EEC [15]. MetS may have a long-term adverse effect on treatment outcomes.

Several studies demonstrated the adverse effects of HUA on digestive and urinary tract tumors [34,35]. Zhang et al. identified a potential link between high UA and EC [7]. This study found that HUA was an independent risk factor during fertility-sparing treatment in patients with EAH/EEC. The patients with HUA had a lower cumulative 32-week CR rate and longer treatment duration. Several studies suggested that soluble UA and urate crystals activate the NLRP3 inflammasome and mediate inflammatory responses by multiple signaling pathways [36,37]. The serum UA also regulated the IR status through the NLRP3 inflammasome [38]. Hu et al. suggested that UA was an independent risk factor for the onset of insulin-insensitive diabetes [39]. We speculated that IR and chronic inflammation mediated by HUA might be a biological mechanism for progesterone resistance in the endometrium, resulting in poor efficacy in patients with EAH/EEC.

In patients with EAH/EEC, HUA showed a strong association with obesity. We further investigated the joint effects of HUA and obesity on treatment outcomes. The cumulative CR rate was significantly lower in patients with both HUA and obesity, suggesting that both played a synergistic role in progestin therapy. The potential mechanism may be correlated to a mutual relationship between obesity/HUA and adiponectin. Leptin, which is negatively associated with obesity [23], may lead to high serum UA levels by impairing the renal excretion of UA and downregulating the expression of hepatic xanthine oxidoreductase [40]. Xanthine oxidoreductase is a rate-limiting enzyme of purine metabolism and is associated with serum UA levels. Battelli et al. found that decreased xanthine oxidoreductase levels led to the proliferation and metastasis of tumor cells in breast, ovarian, gastric, and colorectal cancer [41]. We speculated that the joint regulation of adiponectin mediated by obesity and HUA promoted the inflammatory microenvironment of the endometrium, reducing the efficacy of fertility-sparing treatment in patients with EAH/EEC.

### 4.3. Strengths and Limitations

To the best of our knowledge, the present study is the first to suggest that underweight status has an adverse impact on fertility-sparing therapy in patients with EAH/EEC. The optimal treatment efficacy was achieved at a BMI of 21–22 kg/m^2^. We also found for the first time that HUA was an independent risk factor for conservative treatment efficacy for EAH/EEC. This study also had certain limitations. Firstly, the number of underweight patients in the study population was only 14 (4.9%), which may affect the results of statistical analysis. However, considering the rarity of underweight status in patients with EAH or EEC, our conclusions could be reasonably explained by several potential mechanisms mentioned in this article. Secondly, the analysis was limited by the retrospective nature and restricted follow-up duration of this study, especially for our conclusions on recurrence and reproductive outcomes. Thirdly, we did not assess the effect of weight change during the fertility-sparing treatment upon oncological and reproductive outcomes. The prognostic significance remains to be further studied. In addition, all the patients in our study were Asians of Chinese nationality, so caution should be exercised in applying our findings to other races or groups.

### 4.4. Implications for Clinical Practice and Future Research

Our results indicate that a target BMI interval for weight management should be appropriately established for patients with EAH/EEC. According to the Guidelines (2013) for Managing Overweight and Obesity in Adults, the most effective approach to weight loss includes three components: a reduced calorie diet, increased physical activity, and behavior changes. For adults with a BMI over 40 kg/m^2^ or those with a BMI over 35 kg/m^2^ who also have obesity-related comorbidities and have no response to other treatments, bariatric surgery may be an appropriate option. Our center is now conducting clinical trials to investigate the effects of weight management and weight management combined with loxenatide for patients with EAH/EC under fertility-sparing treatment. The management of underweight patients should not be ignored. The proper enhancement of muscle mass and the improvement of potential hyperlipidemia may be beneficial to fertility-sparing treatment. Previous studies included the underweight subgroup, which was of high risk, in the normal weight subgroup for survival analysis. This may affect the accuracy of the results to some extent and will need more attention in future studies. Attention should also be paid to regulating obesity-related metabolic disorders, especially IR, PCOS, MetS, and the increasing trend of serum UA. Studies suggested that diabetes was a risk factor for containing occult EC in patients with endometrial hyperplasia [42]. The management of blood glucose levels during conservative treatment should not be neglected either. The long-term adverse effects of HUA and MetS on treatment outcomes were more severe than the short-term effects. Early active interventions, preferably before 32 weeks of treatment, should be provided for patients with these comorbidities to improve treatment efficacy. We have conducted a randomized clinical trial that showed a higher rate of early CR in EAH patients treated with megestrol acetate combined with metformin than megestrol acetate alone [43]. Based on this preliminary study, a prospective trial to confirm the efficacy of progestin treatment combined with deregulation of UA for patients with EAH and EEC is warranted, and related future results may shed light on the effectiveness of these therapies.

## 5. Conclusions

This retrospective analysis of 286 patients with EAH/EEC receiving fertility-sparing treatment showed a parabola-shaped relationship between BMI and oncological outcomes. Underweight and overweight/obese status were each associated with lower CR rates and longer treatment duration. Furthermore, we demonstrated that HUA was an independent risk factor for long-term conservative treatment outcomes. Patients exhibiting both obesity and HUA had a worse treatment efficacy compared with obesity or HUA alone. The current conclusions provide important indications for the guidance of clinical work and the design of clinical trials. High-quality prospective studies with a larger sample size are needed to clarify the effects of different weight statuses and related endocrine metabolic disorders, such as HUA, on fertility-sparing treatment in patients with EAH and EEC.

## Figures and Tables

**Figure 1 cancers-14-05024-f001:**
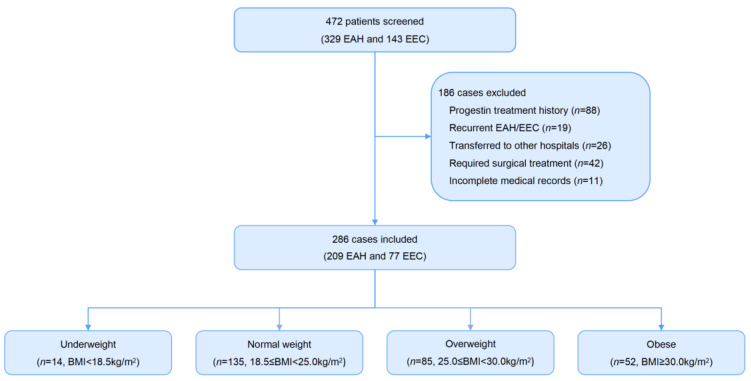
Flow chart of study population selection. EAH, endometrial atypical hyperplasia; EEC, endometrioid endometrial cancer.

**Figure 2 cancers-14-05024-f002:**
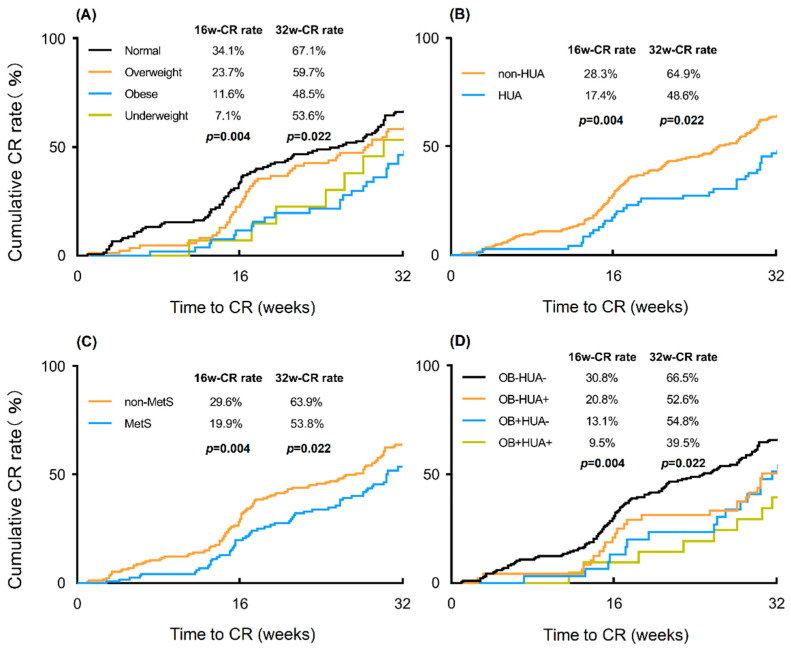
Cumulative CR rate at the 16th and 32nd week of treatment. (**A**) Cumulative CR rate in patients with a different weight status; (**B**) Cumulative CR rate in patients with or without HUA; (**C**) Cumulative CR rate in patients with or without PCOS; (**D**) Cumulative CR rate in patients with or without HUA stratified by obesity. Cumulative CR rate was evaluated by the Kaplan–Meier method and compared by a log-rank test. CR, complete response; HUA, hyperuricemia; PCOS, polycystic ovary syndrome; OB, obesity.

**Figure 3 cancers-14-05024-f003:**
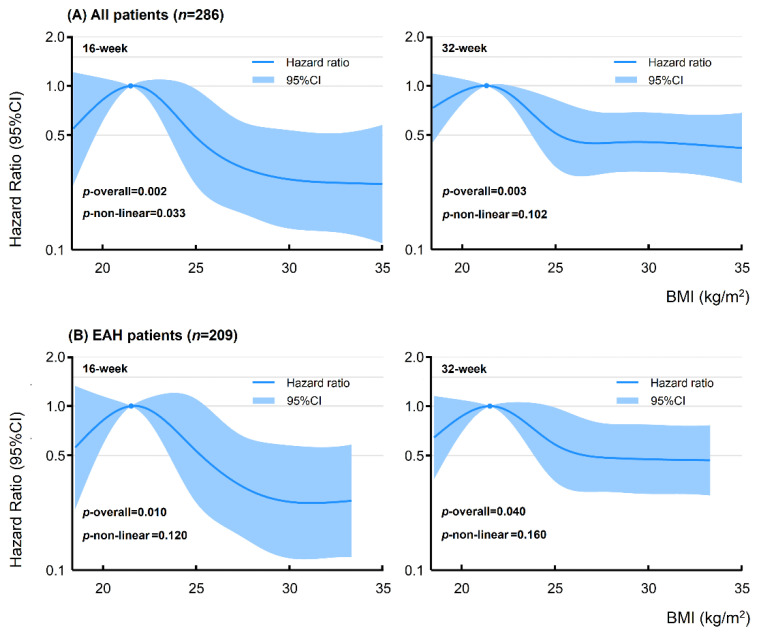
Non-linear analysis between BMI and cumulative CR rate at the 16th and 32nd week of treatment for all patients and the subgroup of patients with EAH. Non-linear analysis was modeled by restricted cubic spline. The HR and 95%CI were evaluated by Cox regression analysis adjusted by age at diagnosis, histology, and progestin therapy. BMI, body mass index; EAH, endometrial atypical hyperplasia; CR, complete response; CI, confidence interval; HR, hazard ratio.

**Figure 4 cancers-14-05024-f004:**
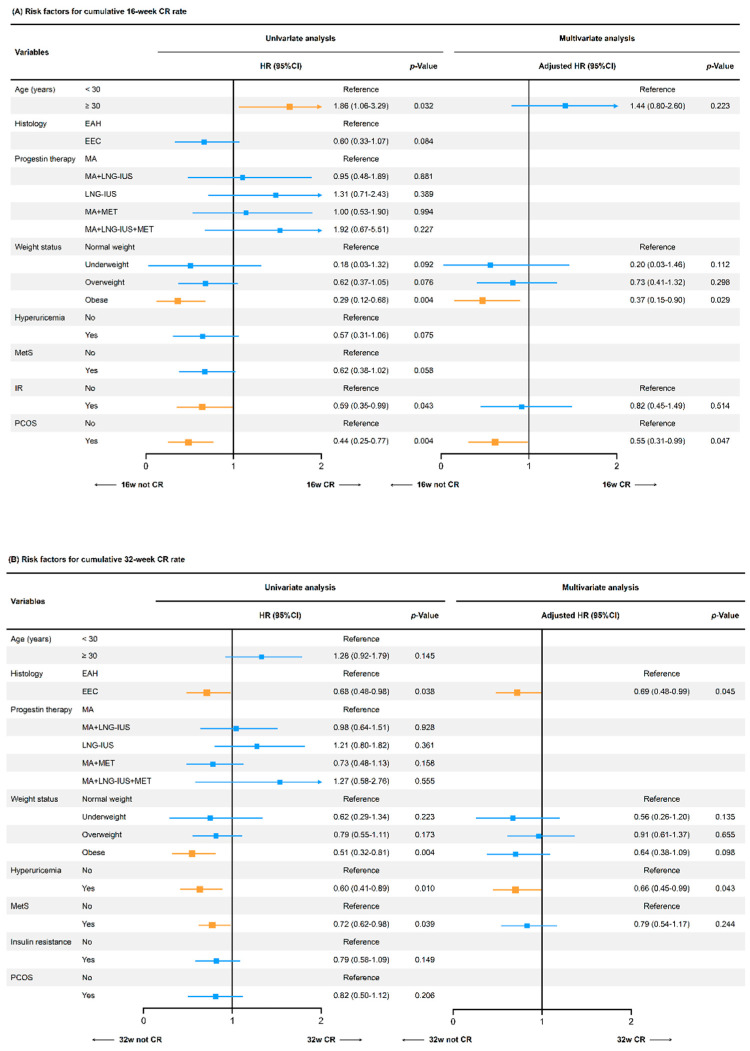
Risk factors associated with the cumulative CR rate at the 16th and 32nd week of treatment. (**A**) Risk factors for the cumulative 16-week CR rate; (**B**) Risk factors for the cumulative 32-week CR rate. The HR and 95%CI were evaluated by univariate or multivariate Cox regression analysis. MA, megestrol acetate; LNG-IUS, levonorgestrel intrauterine device; MET, metformin; IR, insulin resistance; MetS, metabolic syndrome; PCOS, polycystic ovary syndrome; HR, hazard ratio; CI, confidence interval.

**Table 1 cancers-14-05024-t001:** Baseline and metabolic characteristics across weight status.

Variables	Total	Normal Weight (18.5 ≤ BMI < 25 kg/m^2^)	Overweight (25 ≤ BMI < 30 kg/m^2^)	Obese (BMI ≥ 30 kg/m^2^)	Underweight (BMI < 18.5 kg/m^2^)	*p*-Value
No. of patients	286	135 (47.2%)	85 (29.7%)	52 (18.2%)	14 (4.9%)	-
Age (years)	32.0 (29.0–36.0)	32.0 (29.0–36.5)	32.0 (29.0–35.0)	31.0 (28.0–35.0)	33.0 (29.0–37.0)	0.068
Histology						0.951
EAH	209 (73.1%)	98 (72.6%)	64 (75.3%)	37 (71.2%)	10 (71.4%)	
EEC	77 (26.9%)	37 (27.4%)	21 (24.7%)	15 (28.8%)	4 (28.6%)	
Diabetes	32 (11.2%)	5 (3.7%)	15 (17.6%)	12 (23.1%)	0 (0.0%)	<0.001
IR	105 (36.7%)	18 (13.3%)	51 (60.0%)	35 (67.3%)	1 (7.1%)	<0.001
Hyperlipidemia	158 (55.2%)	69 (51.1%)	49 (57.6%)	30 (57.7%)	10 (71.4%)	0.433
Hyper-TC	18 (6.3%)	8 (6.0%)	8 (9.4%)	2 (3.8%)	0 (0.0%)	0.557
Hyper-TG	41 (14.3%)	7 (5.2%)	20 (23.5%)	14 (26.9%)	0 (0.0%)	<0.001
Hypo-HDL	116 (40.6%)	58 (43.0%)	29 (34.1%)	19 (36.5%)	10 (71.4%)	0.053
Hyper-LDL	14 (4.9%)	5 (3.7%)	6 (7.1%)	3 (5.8%)	0 (0.0%)	0.632
MetS	116 (40.6%)	20 (14.8%)	56 (65.9%)	40 (76.9%)	0 (0.0%)	<0.001
Hypertension	98 (34.3%)	31 (23.0%)	40 (47.1%)	26 (50.0%)	1 (7.1%)	<0.001
DOR	63 (22.0%)	36 (26.7%)	18 (22.8%)	7 (14.0%)	2 (14.3%)	0.236
PCOS	104 (36.4%)	37 (27.4%)	35 (41.2%)	28 (53.8%)	4 (28.6%)	0.005
Hyperuricemia	69 (24.1%)	24 (17.8%)	24 (28.2%)	21 (40.4%)	0 (0.0%)	0.001
Progestin therapy						0.128
MA	106 (37.1%)	51 (37.8%)	37 (43.5%)	12 (23.1%)	6 (42.9%)	
MA+LNG-IUS	52 (18.2%)	22 (16.3%)	13 (15.3%)	11 (21.2%)	6 (42.9%)	
LNG-IUS	55 (19.2%)	29 (21.5%)	14 (16.5%)	12 (23.1%)	0 (0.0%)	
MA+MET	63 (22.0%)	30 (22.2%)	16 (18.8%)	15 (28.8%)	2 (14.3%)	
MA+LNG-IUS+MET	10 (3.5%)	3 (2.2%)	5 (5.9%)	2 (3.8%)	0 (0.0%)	
Follow-up (months)	19.1 (12.1–27.8)	21.5 (13.1–28.9)	17.8 (11.1–24.7)	18.5 (11.4–24.8)	20.7 (9.8–28.3)	0.222

Data are shown as median (IQR) or number (%). *p*-value among different weight groups was evaluated by one-way ANOVA, Chi-square, or Fisher’s exact test. BMI, body mass index; IQR, interquartile range; EAH, endometrial atypical hyperplasia; EEC, endometrioid endometrial cancer; IR, insulin resistance; Hyper-TC, hypercholesterolemia; Hyper-TG, hypertriglyceridemia; Hypo-HDL, hypo-high-density lipoprotein cholesterolemia; Hyper-LDL, hyper-low-density lipoprotein cholesterolemia; MetS, metabolic syndrome; DOR, diminished ovarian reserve; PCOS, polycystic ovary syndrome; MA, megestrol acetate; LNG-IUS, levonorgestrel intrauterine device; MET, metformin.

**Table 2 cancers-14-05024-t002:** Correlation between metabolic disorders and weight status.

Variables	Overweight vs. Normal Weight		Obese vs. Normal Weight		Underweight vs. Normal Weight	
	OR (95%CI)	*p*-Value	OR (95%CI)	*p*-Value	OR (95%CI)	*p*-Value
Diabetes	5.79 (2.01–16.74)	0.001	8.00 (2.63–24.26)	<0.001	- *	- *
IR	9.94 (5.10–19.35)	<0.001	13.03 (6.06–28.00)	<0.001	0.52 (0.06–4.28)	0.541
Hyperlipidemia	1.30 (0.75–2.26)	0.346	1.37 (0.71–2.63)	0.370	2.64 (0.77–9.04)	0.122
Hyper-TC	1.79 (0.63–5.06)	0.275	0.74 (0.15–3.67)	0.708	- *	- *
Hyper-TG	6.37 (2.50–16.20)	<0.001	8.17 (2.95–22.60)	<0.001	- *	- *
Hypo-HDL	0.67 (0.38–1.19)	0.174	0.77 (0.39–1.50)	0.435	3.50 (1.02–11.97)	0.047
Hyper-LDL	2.12 (0.64–7.69)	0.212	1.94 (0.43–8.70)	0.388	- *	- *
MetS	12.23 (6.19–24.15)	<0.001	24.06 (9.98–58.02)	<0.001	- *	- *
Hypertension	3.19 (1.75–5.79)	<0.001	3.90 (1.92–7.92)	<0.001	0.30 (0.37–2.46)	0.263
DOR	0.83 (0.42–1.65)	0.603	0.49 (0.20–1.22)	0.124	0.65 (0.13–3.24)	0.601
PCOS	1.71 (0.93–3.15)	0.086	2.89 (1.44–5.82)	0.003	0.67 (0.18–2.54)	0.552
Hyperuricemia	1.90 (0.98–3.70)	0.058	3.06 (1.50–6.25)	0.002	- *	- *

Data are shown as OR (95%CI). OR and 95%CI were evaluated by logistic regression analysis adjusted for age at diagnosis and histology. * Data were unavailable for statistical analysis because no cases of metabolic abnormalities were observed in underweight status. OR, odds ratio; CI, confidence interval; IR, insulin resistance; Hyper-TC, hypercholesterolemia; Hyper-TG, hypertriglyceridemia; Hypo-HDL, hypo-high-density lipoprotein cholesterolemia; Hyper-LDL, hyper-low-density lipoprotein cholesterolemia; MetS, metabolic syndrome; DOR, diminished ovarian reserve; PCOS, polycystic ovary syndrome.

**Table 3 cancers-14-05024-t003:** Fertility-sparing treatment outcomes across weight status.

Variables	Total	Normal Weight (18.5 ≤ BMI < 25 kg/m^2^)	Overweight (25 ≤ BMI < 30 kg/m^2^)	Obese (BMI ≥ 30 kg/m^2^)	Underweight (BMI < 18.5 kg/m^2^)	*p*-Value
No. of patients	286	135 (47.2%)	85 (29.7%)	52 (18.2%)	14 (4.9%)	-
CR rate						
Till 16-week	25.7%	34.1%	23.7%	11.6%	7.1%	0.004
Till 32-week	60.9%	67.1%	59.7%	48.5%	53.6%	0.022
Till 24-month	98.2%	97.1%	100.0%	100.0%	100.0%	0.098
Time to CR (mo)	6.2 (3.6–8.7)	5.1 (3.3–7.9)	5.9 (3.6–8.9)	7.4 (5.7–11.9)	6.8 (4.9–9.5)	0.015
Relapse rate ^a^	29.3%	29.6%	37.6%	7.3%	0.0%	0.418
Time to relapse (mo)	13.8 (8.6–19.8)	12.6 (9.6–22.2)	14.3 (8.8–22.5)	6.3 (6.1–6.5)	- *	0.796
Pregnant ^b^	36 (22.4%)	17 (21.8%)	10 (20.4%)	6 (22.2%)	3 (42.9%)	0.610
Miscarriage ^c^	8 (22.2%)	5 (29.4%)	1 (10.0%)	2 (33.3%)	0 (0.0%)	0.565
Live birth ^c^	18 (50.0%)	8 (47.1%)	6 (60.0%)	2 (33.3%)	2 (66.7%)	0.737

Data are shown as median (IQR) or number (%). Cumulative CR/relapse rates were evaluated by Kaplan–Meier method and compared by log-rank test. ^a^ Relapse rate among patients who achieved CR till the 24-month. ^b^ Pregnant rate among patients who attempted pregnancy after achieving CR. ^c^ Miscarriage and live birth rate among patients who achieved pregnancy. * Data were unavailable for statistical analysis because no relapse cases were observed in underweight status. BMI, body mass index; IQR, interquartile range; CR, complete response.

**Table 4 cancers-14-05024-t004:** Metabolic factors associated with fertility-sparing treatment outcomes.

Variables		No. of Patients	16-Week CR Rate	*p*-Value	32-Week CR Rate	*p*-Value	Time to CR (mo)	*p*-Value
Diabetes	No	254 (88.8%)	26.2%	0.554	62.3%	0.214	6.0 (3.6–8.5)	0.045
	Yes	32 (11.2%)	21.9%		50.7%		7.1 (3.7–12.0)	
IR	No	181 (63.3%)	29.5%	0.041	62.8%	0.147	5.8 (3.4–8.4)	0.073
	Yes	105 (36.7%)	19.0%		57.8%		6.7 (3.9–9.5)	
Hyperlipidemia	No	128 (44.8%)	27.0%	0.594	63.4%	0.391	5.8 (3.6–9.4)	0.936
	Yes	158 (55.2%)	24.5%		58.8%		6.6 (3.6–8.5)	
Hyper-TC	No	268 (93.7)	26.6%	0.158	61.9%	0.261	6.2 (3.5–8.9)	0.772
	Yes	18 (6.3%)	11.5%		46.9%		6.4 (4.0–8.0)	
Hyper-TG	No	245 (85.7%)	27.5%	0.093	62.3%	0.172	6.1 (3.5–8.9)	0.492
	Yes	41 (14.3%)	14.7%		53.0%		7.1 (4.0–8.5)	
Hypo-HDL	No	170 (59.4%)	24.9%	0.615	61.8%	0.802	6.0 (3.7–8.6)	0.847
	Yes	116 (40.6%)	26.8%		59.8%		6.6 (3.5–8.8)	
Hyper-LDL	No	272 (95.1%)	25.8%	0.722	62.1%	0.148	6.1 (3.6–8.8)	0.507
	Yes	14 (4.9%)	22.1%		37.7%		7.5 (3.6–8.6)	
MetS	No	170 (59.4%)	29.6%	0.055	63.9%	0.037	5.9 (3.4–8.2)	0.056
	Yes	116 (40.6%)	19.9%		53.8%		6.7 (3.9–9.6)	
Hypertension	No	188 (65.7%)	28.4%	0.136	62.0%	0.182	6.0 (3.5–8.9)	0.560
	Yes	98 (34.3%)	20.5%		53.5%		6.8 (3.8–8.5)	
DOR	No	223 (78.0%)	23.4%	0.147	58.7%	0.199	6.6 (3.6–9.1)	0.052
	Yes	63 (22.0%)	34.1%		63.5%		4.9 (3.5–7.8)	
PCOS	No	182 (63.6%)	31.6%	0.003	61.0%	0.203	5.4 (3.4–8.5)	0.045
	Yes	104 (36.4%)	15.4%		57.7%		6.9 (4.1–9.4)	
Hyperuricemia	No	217 (75.9%)	28.3%	0.071	64.9%	0.009	5.8 (3.5–8.0)	0.047
	Yes	69 (24.1%)	17.4%		48.6%		7.1 (4.0–10.9)	

Data are shown as median (IQR) or number (%). Cumulative CR rates were evaluated by Kaplan–Meier method and compared by log-rank test. IQR, interquartile range; CR, complete response; IR, insulin resistance; Hyper-TC, hypercholesterolemia; Hyper-TG, hypertriglyceridemia; Hypo-HDL, hypo-high-density lipoprotein cholesterolemia; Hyper-LDL, hyper-low-density lipoprotein cholesterolemia; MetS, metabolic syndrome; DOR, diminished ovarian reserve; PCOS, polycystic ovary syndrome.

## Data Availability

The datasets generated during and/or analyzed during the current study are not publicly available but are available from the corresponding author upon reasonable request.

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
