# Peer review of "Effects of Weight Status and Related Metabolic Disorders on Fertility-Sparing Treatment Outcomes in Endometrial Atypical Hyperplasia and Endometrial Cancer: A Retrospective Study"

_cancers, 2022, doi:10.3390/cancers14205024_

Round 1
Reviewer 1 Report
1. As you mentioned that some patients also received oral metformin at a dose of 1500 mg/d, depending on medical complications, does the use of metformin affect the prognosis?
2. As we know weight control is also an important issue during treatment. Did you trace the body weight change of the patients during the treatment and follow up?
Author Response
Response to Reviewer 1 Comments
Manuscript ID: cancers-1938448
Title of Paper: Effects of Weight Status and Related Metabolic Disorders on Fertility-sparing Treatment Outcomes in Endometrial Atypical Hyperplasia and Endometrial Cancer: A Retrospective Study
Response: We’d like to extend our sincere gratitude for your patience and dedication to reviewing the manuscript. After revising our manuscript according to the given comments, we herein provide our point-by-point responses to the reviewer’s comments with explanations of the corrections/additions. All changes are in red font in the response letter and revised manuscript. We hope that the revised manuscript is suitable for publication in Cancers. We welcome additional comments and suggestions.
Point 1: As you mentioned that some patients also received oral metformin at a dose of 1500 mg/d, depending on medical complications, does the use of metformin affect the prognosis?
Response 1: Thank you very much for your careful review and professional comments. Regarding the effects of metformin on the treatment prognosis, we previously conducted a randomized controlled study (PMID: 31961463). Results showed that for EAH patients, the early CR rate (cumulative CR rate till 16-week) might be improved by adding metformin into megestrol acetate (MA) therapy. However, no significant difference was found in the long-term treatment outcome (cumulative CR rate till 30-week). While for EEC patients, metformin did not show a significant improvement. Phase III trials including a sufficient number of EEC patients were needed to validate further the effect of metformin. These conclusions were published in BJOG, 2020 (PMID: 31961463). .
Metformin might indeed be one of the confounders in the statistical analysis. To address this issue, the following improvements were made to confirm and strengthen the reliability of our study:
- In total, 73 of 286 cases (25.5%) received an initial treatment containing metformin. We used Fisher's exact test to compare the use of metformin across weight status. No significant difference in distribution was found (p=0.391).
- We analyzed the effects of metformin on fertility-sparing treatment outcomes using Kalpan-Meier survival curves. Results showed that the improvement of 16-week CR rate and 32-week CR rate by metformin did not reach statistically significant (p=0.870, p=0.136). However, this might be due to the fact that we did not perform a subgroup analysis of the study population base on histology.
- We always regarded the therapy regimens (progestin with or without metformin) as one of the important confounders, and included it in the Cox regression model for adjustment. In contrast to MA alone, none of the other treatment regimens (MA+MET, LNG-LUS, MA+LNG-LUS, MA+LNG-IUS+MET) was a protective or risk factor for prognosis. On this basis, we believed that the potential prognostic impact of metformin in this study did not prevent us from reaching reliable conclusions.
Point 2: As we know weight control is also an important issue during treatment. Did you trace the body weight change of the patients during the treatment and follow up?
Response 2: Thank you very much for your careful review, this is an issue that deserves to be explored in depth. We performed comprehensive evaluations of all EAH/EEC patients before giving initial treatment, which included measurements of weight, BMI, waist circumference, body fat percentage, skeletal muscle content, VFA, and some other indicators. During the fertility-sparing treatment, patients underwent hysteroscopy every 3 months to evaluate the treatment efficacy. We measured the above-mentioned indicators again to trace their changes before each hysteroscopy. After achieving CR, we recommended patients endometrial aspiration biopsy (or hysteroscopy if necessary) at least every 6 months, which was also an appropriate chance for us to follow up on the weight changes.
Previously, overweight and obese patients lost weight primarily through self-management. Since 2020 we have been providing patients with scientific weight management and now developed a more systematic clinical treatment strategy. Therapeutic lifestyle changes (TLC) were our preferred option to promote weight loss through reduced calorie diet, increased physical activity and behavior changes. If the desired goal (weight loss ≥5%) was not achieved within a reasonable time (3-6 months), metformin, orlistat, liraglutide, or other medications would be considered to assist in weight management as appropriate.
We are conducting a retrospective analysis of the effects of weight management on fertility-sparing treatment outcomes and metabolic profiles in EAH and early EEC patients. If reliable conclusions can eventually be drawn, we would be happy to share them with you.

Reviewer 2 Report
Nice written, clear overview of the research with interessting results. Few small points to consider.
121. what is ment with pathological improvement? Please elaborate for more clear clinical application.
213. For obese patients. the cumulative relapse rate seems very low, but the time to relapse seems very short. It hints to a significant difference comparing to the other groups combined. It could be worth exploring.
276 what pathology type of ECC is ment? this is not explained in the methods. You should only have patients with eiter grade 1 EEC. I believe you mean atypical hyperplasia vs endometrioid carcinoma.
Author Response
Response to Reviewer 2 Comments
Manuscript ID: cancers-1938448
Title of Paper: Effects of Weight Status and Related Metabolic Disorders on Fertility-sparing Treatment Outcomes in Endometrial Atypical Hyperplasia and Endometrial Cancer: A Retrospective Study
Response: We’d like to extend our sincere gratitude for your patience and dedication to reviewing the manuscript. After revising our manuscript according to the given comments, we herein provide our point-by-point responses to the reviewer’s comments with explanations of the corrections/additions. All changes are in red font in the response letter and revised manuscript. We hope that the revised manuscript is suitable for publication in Cancers. We welcome additional comments and suggestions.
Point 1: Nice written, clear overview of the research with interesting results. Few small points to consider
Response 1: Thank you so much for your positive expectation.
Point 2: 121. what is ment with pathological improvement? Please elaborate for more clear clinical application.
Response 2: Thank you very much for your careful review and professional comments. The term "pathological improvement" in this manuscript meant that the endometrial specimens obtained through hysteroscopy were diagnosed as particle response by pathologists, i.e., the presence of complex/simple hyperplasia lesions in EAH patients after treatment, or atypical/complex/simple hyperplasia lesions in EEC patients after treatment. Based on your suggestion, we have interpreted the assessment of treatment efficacy more precisely in the revised version as follows:
Materials and Methods, Page 3, Line 124-131: “Complete response (CR) was defined as an absence of hyperplasia or cancer. The pathology of endometrial specimens could be secretory/proliferative endometrium after treatment. Partial response was defined as a pathological improvement, i.e., the presence of complex/simple hyperplasia lesions in endometrial specimens of EAH patients after treatment, or atypical/complex/simple hyperplasia lesions in EEC patients after treatment. Disease progression was defined as the possibility of EC in EAH or the presence of myometrial infiltration or extra-uterine metastasis in patients with EEC. “
Point 3: 213. For obese patients. the cumulative relapse rate seems very low, but the time to relapse seems very short. It hints to a significant difference comparing to the other groups combined. It could be worth exploring.
Response 3: We are also very interested in the topic you raised. Previously, we held the same doubts and conducted further discussions. Pairwise comparisons were performed to analyze the cumulative relapse rate and time to relapse across weight status. The p-values for pairwise comparisons were adjusted by Bonferroni. Results showed that although relapse rate was lower in obese patients, no differences reached statistical significance. Surprisingly, time to relapse was significantly shorter in obese patients when compared to normal weight patients (6.3 months vs. 12.6 months, p=0.031). Due to the limited sample size (only 2 obese patients presented with recurrence) and follow-up time, we could not obtain more reliable conclusions. Your interest in this topic is much appreciated and we have added this section in the revised version to throw light on:
Results, Page 7, Line 249-252: “However, obese cases seemed to show a lower relapsed rate (7.3%) and a shorter time to relapse (6.3±0.2 months). In pairwise comparison, the recurrence of obese patients occurred significantly sooner than normal weight patients (6.3 months vs 12.6 months, p=0.031).”
A large-sample retrospective analysis is underway in our center, focusing on the factors that affect relapse in EAH or EEC patients who received fertility-sparing treatment. If reliable conclusions can eventually be drawn, we would be happy to share them with you.
Point 4: 276 what pathology type of ECC is ment? this is not explained in the methods. You should only have patients with eiter grade 1 EEC. I believe you mean atypical hyperplasia vs endometrioid carcinoma.
Response 4: Thank you very much for pointing out the spelling errors. The whole manuscript has been rechecked and corrected carefully to ensure that we describe our work more clearly and precisely in the revised version.

Reviewer 3 Report
This is an interesting retrospective study which aimed to evaluate the effects of different weight status and related metabolic disorders on the oncological and reproductive outcomes of fertility-sparing treatment in Atypical Endometrial Hyperplasia (AEH) and Early Stage Endometrioid Endometrial Cancer (EEC). In fact, fertility sparing treatment (FST) of EC is a crucial topic in literature to date and this article represents a valid contribution to the field, since it analyzes the value in response rate related to metabolic disorders and obesity, which is a matter of debate in literature.
I have the following comments to the Authors:
· English language needs to be checked throughout the whole manuscript (minor spelling mistakes).
· Material and Methods: in order to make the study reproducible, Authors should explain what they mean for “expert gynecologic pathologists”
· Material and Methods: in order to make the study reproducible, Authors should be more specific about the choice of progestin therapy in each patient: they described 3 approaches, however they did not mention any criteria for applying each regimen to a patient, if any.
· Methods: In order to make the study reproducible, Authors should describe in details how were selected the patients included in the study and how was selection bias excluded during this phase.
· Discussion: Authors should include a section in the discussion describing strengths and limitations of this study.
· Discussion: In this article, Authors analyzed the effects of different weight status and related metabolic disorders on the oncological and reproductive outcomes of fertility-sparing treatment in AEH and EEC. In this sense, endometrial hyperplasia in patients with diabetes mellitus can hide a certain risk of containing an occult endometrial carcinoma in some cases, which should be mentioned in the text in order to make the article more modern and praiseworthy (PMID: 31203571).
Author Response
Response to Reviewer 3 Comments
Manuscript ID: cancers-1938448
Title of Paper: Effects of Weight Status and Related Metabolic Disorders on Fertility-sparing Treatment Outcomes in Endometrial Atypical Hyperplasia and Endometrial Cancer: A Retrospective Study
Response: We’d like to extend our sincere gratitude for your patience and dedication to reviewing the manuscript. After revising our manuscript according to the given comments, we herein provide our point-by-point responses to the reviewer’s comments with explanations of the corrections/additions. All changes are in red font in the response letter and revised manuscript. We hope that the revised manuscript is suitable for publication in Cancers. We welcome additional comments and suggestions.
Point 1: This is an interesting retrospective study which aimed to evaluate the effects of different weight status and related metabolic disorders on the oncological and reproductive outcomes of fertility-sparing treatment in Atypical Endometrial Hyperplasia (AEH) and Early Stage Endometrioid Endometrial Cancer (EEC). In fact, fertility sparing treatment (FST) of EC is a crucial topic in literature to date and this article represents a valid contribution to the field, since it analyzes the value in response rate related to metabolic disorders and obesity, which is a matter of debate in literature. I have the following comments to the Authors:
Response 1: Thank you so much for your positive expectation.
Point 2: English language needs to be checked throughout the whole manuscript (minor spelling mistakes).
Response 2: We sincerely value your suggestion. The language of our manuscript has been checked and corrected carefully to ensure that we describe our work more clearly and precisely in the revised version.
Point 3: Material and Methods: in order to make the study reproducible, Authors should explain what they mean for “expert gynecologic pathologists”.
Response 3: Thank you very much for your professional comments. The term "expert gynecologic pathologist" refers to a specialist with extensive practical experience in the field of gynecologic tumor pathology diagnosis and with the title of the associate chief physician or above. For example, Dr. Yiqin Wang, MD, Ph.D., has been engaged in gynecological pathology diagnosis and basic research for more than 10 years at Obstetrics and Gynecology Hospital of Fudan University. We have added the explanation of "expert gynecologic pathologist" in the revised version and expressed our sincere gratitude to Dr. Yiqin Wang for her assistance and guidance in Acknowledgments:
Materials and Methods, Page 2, Line 70-76: “Following the World Health Organization (WHO) pathological classification (2014), the diagnosis of EEC or EAH (equivalent to endometrial intraepithelial neoplasia, which was utilized more frequently in WHO 2020) was confirmed by two experienced gynecological pathologists. Both of them were with extensive practice experience in the field of gynecologic tumor pathology diagnosis and with their titles of the associate chief physician or above. “
Acknowledgments, Page 14, Line 477-479: “We appreciate Dr. Yiqin Wang for providing us expert suggestions in the field of pathology and Chen Huang’s contribution to statistical guidance. Furthermore, we thank all the patients who participated in this study. “
Point 4: Material and Methods: in order to make the study reproducible, Authors should be more specific about the choice of progestin therapy in each patient: they described 3 approaches, however they did not mention any criteria for applying each regimen to a patient, if any.
Response 4: Thank you very much for your careful review. In this retrospective analysis, patients were mainly given the corresponding progestin therapy after random allocation based on the requirements of clinical trial in which they participated (NCT01968317, NCT03241914, and NCT03241888). Therefore, we only presented the implementation of the 3 progestin therapies (MA; LNG-IUS; MA+LNG-IUS) in Materials and Methods section, instead of the screening and regimens of all clinical trials. The heterogeneity of progestin therapy might confound the study results. To address this issue, we took the following improvements to enhance the reliability of the study:
- In Table 1 we used Fisher's exact test to compare the distribution of different progestin therapy across weight status. No significant differences were found (p=0.128; Table1).
- We always regarded progestin therapy as one of the important confounders, and included it in the Cox regression model for adjustment. In contrast to MA alone, none of the other treatment regimens was a protective or risk factor for prognosis.
Point 5: Methods: In order to make the study reproducible, Authors should describe in details how were selected the patients included in the study and how was selection bias excluded during this phase.
Response 5: Thank you very much for your careful review and professional advice. We screened patients with EAH and early-stage EEC eligible for fertility-sparing treatment in strict accordance with the NCCN guidelines. After being fully informed that the preferred option was hysterectomy and the associated risks of conservative treatment, patients were offered appropriate progestin therapy. A total of 472 patients (including 209 with EAH and 77 with early EEC) were included in this retrospective study to assess the effects of weight status and metabolic abnormalities on fertility-sparing treatment outcomes in EAH and EEC. To reduce the influence of confounding factors, we excluded patients with the following conditions: a history of progestin therapy; recurrent cases; and those who were transferred to another hospital or required surgery before the first assessment of treatment efficacy. Figure 1 illustrated the screening process for the study population. We have added details of the 186 excluded patients in the Results section as follows:
Results, Page 4, Line 163-171: " 186 patients were excluded, including 88 cases who had accepted progestin therapy for more than one month before the first endometrial assessment at our center. We did not analyze these patients because of the potential effects of their previous treatment (including cyclic progestin, continued progestin, oral contraceptive pills, or LNG-IUS), particularly those who had poorer outcomes in initial hospitals. Before the first hysteroscopy was performed to assess the conservative efficacy, 42 cases requested surgical treatment, and 26 cases requested transfer to another hospital were excluded. Besides, 19 cases with recurrent EAH or EEC, and 11 cases with incomplete clinical data were also excluded. “
Patients were mainly given the corresponding progestin therapy after random allocation based on the requirements of clinical trial in which they participated. In our response to Point 4, we analyzed how to control for this possible selection bias. We hope that our explanation will make the article more reproducible and reliable.
Point 6: Discussion: Authors should include a section in the discussion describing strengths and limitations of this study.
Response 6: Thank you for your suggestion We have analyzed the strengths and limitations of this retrospective study in the Discussion section of the manuscript. Look forward to your professional comments.
Discussion, Page 13, Line 404-419: “To the best of our knowledge, the present study is the first to suggest that underweight status has an adverse impact on fertility-sparing therapy in patients with EAH/EEC. The optimal treatment efficacy was achieved at a BMI of 21–22 kg/m2. We also found for the first time that HUA was an independent risk factor for conservative treatment efficacy for EAH/EEC. This study also had certain limitations. Firstly, the number of underweight patients in the study population was only 14 (4.9%), which may affect the results of statistical analysis. However, considering the rarity of underweight status in patients with EAH or EEC, our conclusions could be reasonably explained by several potential mechanisms mentioned in this article. Secondly, the analysis was limited by the retrospective nature and restricted follow-up duration of this study, especially for our conclusions on recurrence and reproductive outcomes. Thirdly, we did not assess the effect of weight change during the fertility-sparing treatment upon oncological and reproductive outcomes The prognostic significance remains to be further studied. In addition, all the patients in our study were Asians of Chinese nationality, so caution should be exercised in applying our findings to other races or groups. “
Point 7: Discussion: In this article, Authors analyzed the effects of different weight status and related metabolic disorders on the oncological and reproductive outcomes of fertility-sparing treatment in AEH and EEC. In this sense, endometrial hyperplasia in patients with diabetes mellitus can hide a certain risk of containing an occult endometrial carcinoma in some cases, which should be mentioned in the text in order to make the article more modern and praiseworthy (PMID: 31203571).
Response 7: Thank you very much for your suggestion. We have carefully read this meta-analysis (PMID:31203571), which showed that diabetes was a risk factor for combined occult EC in women diagnosed with endometrial hyperplasia (OR=1.96; 95% CI, 1.07-3.60; P=0.03). It was recommended that patients should manage their blood glucose during the conservative treatment in order to prevent the progression of the disease. This is such a meaningful conclusion that we have added the conclusions and cited this paper in the revised manuscript:
Discussion, Page 13, Line 436-438: " Studies suggested that diabetes was a risk factor for containing occult EC in patients with endometrial hyperplasia [42]. The management of blood glucose levels during conservative treatment should not be neglected either."

Reviewer 4 Report
The research is well-written and elucidates the correlation between a patient's BMI and the efficacy of conservative therapy of endometrial lesions. This reviewer brought up only a few minor points.
#1. When was the BMI of the patients calculated, and did it alter during treatment? If they did fluctuate during therapy, for instance, if a patient with obesity reduced her weight, would the period of treatment to CR be decreased? Coudl you clarify this point? This seems to be an essential practical consideration about how the weight of patients should be managed during treatment.
#2. Additionally, completion rates for hormone therapy should be included. Can we assume that treatment adherence is comparable between different BMI groups? Obese individuals frequently struggle with self-management, and adherence to drugs and various therapy may vary among them. If treatment adherence is demonstrably worse in the overweight and obese groups, the difference in time to CR cannot be attributable solely to BMI.
#3. In the latest WHO classification (WHO 2020), EIN is utilized more frequently than EAH. Include commentary on whether the EAH applied in this study may be considered equivalent to EIN. Because this is essential for the generalizability of this study's findings.
Author Response
Response to Reviewer 4 Comments
Manuscript ID: cancers-1938448
Title of Paper: Effects of Weight Status and Related Metabolic Disorders on Fertility-sparing Treatment Outcomes in Endometrial Atypical Hyperplasia and Endometrial Cancer: A Retrospective Study
Response: We’d like to extend our sincere gratitude for your patience and dedication to reviewing the manuscript. After revising our manuscript according to the given comments, we herein provide our point-by-point responses to the reviewer’s comments with explanations of the corrections/additions. All changes are in red font in the response letter and revised manuscript. We hope that the revised manuscript is suitable for publication in Cancers. We welcome additional comments and suggestions.
Point 1: The research is well-written and elucidates the correlation between a patient's BMI and the efficacy of conservative therapy of endometrial lesions. This reviewer brought up only a few minor points.
Response 1: Thank you so much for your positive expectation.
Point 2: When was the BMI of the patients calculated, and did it alter during treatment? If they did fluctuate during therapy, for instance, if a patient with obesity reduced her weight, would the period of treatment to CR be decreased? Could you clarify this point? This seems to be an essential practical consideration about how the weight of patients should be managed during treatment.
Response 2: Thank you very much for your careful review and professional comments. We performed comprehensive evaluation of all EAH/EEC patients before giving initial treatment, which included measurements of weight, BMI, waist circumference, body fat percentage, skeletal muscle content, VFA, and some other indicators. During the fertility-sparing treatment, patients underwent hysteroscopy every 3 months to evaluate the treatment efficacy. We measured the above-mentioned indicators again to trace their changes before each hysteroscopy. After achieving CR, we recommended patients endometrial aspiration biopsy (or hysteroscopy if necessary) at least every 6 months, which was also an appropriate chance for us to follow up on the weight changes. Despite our long-term follow-up of patients' BMI, we failed to explore in depth the potential effects of weight change on fertility-sparing treatment. In the Discussion section (Discussion, Page 13, Line 415-417), we acknowledged this limitation: “We did not assess the effect of weight change during the fertility-sparing treatment upon oncological and reproductive outcomes. The prognostic significance remains to be further studied.”
In response to your question, we fully agree that weight change may be one of the confounders in conservation treatment. In order to make the study results more reliable, we offer the following explanation:
- Previously, overweight and obese patients lost weight primarily through self-management which had little effect. Less than 10% of patients lose ≥5% of their body weight during the fertility-sparing treatment. In a preliminary analysis conducted on the above basis, we did not find significant effects of weight management on CR rate, time to CR, or recurrence of fertility-sparing treatment.
- Since 2020 we have been providing patients with scientific weight management and now developed a more systematic clinical treatment strategy. Therapeutic lifestyle changes (TLC) were our preferred option to promote weight loss through a reduced calorie diet, increased physical activity, and behavior changes. If the desired goal (weight loss ≥5%) was not achieved within a reasonable time (3-6 months), metformin, orlistat, liraglutide, or other medications would be considered to assist in weight management as appropriate.
We are conducting a retrospective analysis of the effects of weight management on fertility-sparing treatment outcomes and metabolic profiles in EAH and early EEC patients. Preliminary analyses showed that for overweight/obese patients, weight loss of 5-10% would improve fertility-sparing treatment outcomes. Weight loss ≥10% would significantly improve CR rate and shorten the time to CR. A previous study showed that weight loss ≥5% could significantly improve the pregnancy rate (88.0% vs. 36.8%) and live birth rate (64.0% vs. 34.6%) (PMID: 34285588). Therefore, we speculated that the oncologic and reproductive outcomes might be significantly improved if EAH/EEC patients could have their weight loss ≥5% through TLC or other modalities while undergoing fertility-sparing treatment. If reliable conclusions can eventually be drawn, we would be happy to share them with you.
Point 3: Additionally, completion rates for hormone therapy should be included. Can we assume that treatment adherence is comparable between different BMI groups? Obese individuals frequently struggle with self-management, and adherence to drugs and various therapy may vary among them. If treatment adherence is demonstrably worse in the overweight and obese groups, the difference in time to CR cannot be attributable solely to BMI.
Response 3: Thank you very much for your valuable suggestions. Before giving the initial treatment, we established online contact with all EAH/EEC patients to facilitate management. During the fertility-sparing treatment, we actively followed up with patients on their medications and ensured their timely completion of hysteroscopy to assess the treatment efficacy. The advantages of this model were: 1). convenient and fast contact with patients allowed us to keep track of their medication compliance, reducing the rate of unnecessary lost follow-up; 2) if patients had any medication difficulties or adverse reactions, they could communicate with us at any time and adjust the medication regimens if necessary.
Thanks to our close contact with patients and their strong demand for fertility, no cases were found to behave significantly poorer compliance with progestin therapy during the treatment and follow-up. Therefore, we could basically assume that the compliance of patients with different weight status was essentially the same. We have added the description of patient compliance with fertility-sparing treatment in the revised version to make our results more reliable:
Materials and Methods, Page 3, Line 116-118: “During the fertility-sparing treatment, all patients were actively managed in order to keep track of their medication adherence and adverse effects.”
Results, Page 4, Line 176-177:” As of the last follow-up, no patients were found to have poor medication adherence. “
Point 4: In the latest WHO classification (WHO 2020), EIN is utilized more frequently than EAH. Include commentary on whether the EAH applied in this study may be considered equivalent to EIN. Because this is essential for the generalizability of this study's findings.
Response 4: Thanks for your professional suggestions. The terminology mentioned in this study was mainly based on the histological classification of uterine tumors in WHO 2014. After receiving your suggestion, we carefully reviewed the latest classification in WHO 2020 and added the description of endometrial intraepithelial neoplasia (EIN) in the revised manuscript:
Materials and Methods, Page 2, Line 70-76: “Following the World Health Organization (WHO) pathological classification (2014), the diagnosis of EEC or EAH (equivalent to endometrial intraepithelial neoplasia, which was utilized more frequently in WHO 2020) was confirmed by two experienced gynecological pathologists. Both of them were with extensive practice experience in the field of gynecologic tumor pathology diagnosis and with their titles of the associate chief physician or above. “
